# Exploring the effects of habitat management on grassland biodiversity: A case study from northern Serbia

**Dubravka Milić** [1]*, **Milica Rat**[1], **Bojana Bokić** [1], **Sonja Mudri-Stojnić**[1],
**Nemanja Milošević** [2], **Nataša Sukur**[2], **Dušan Jakovetić**[2], **Boris Radak**[1], **Tamara Tot**[1],
**Dušanka Vujanović** [3], **Goran Anačkov**[1], **Dimitrije Radišić** [1]

1 Department of Biology and Ecology, Faculty of Sciences, University of Novi Sad, Novi Sad, Serbia,
2 Department of Mathematics and Informatics, Faculty of Sciences, University of Novi Sad, Novi Sad, Serbia,
3 BioSense Institute, University of Novi Sad, Novi Sad, Serbia

* dubravka.milic@dbe.uns.ac.rs

**Data Availability Statement:** All relevant data are within the manuscript and its Supporting Information files.

## Abstract

Grasslands represent a biodiversity hotspot in the European agricultural landscape, their restoration is necessary and offers a great opportunity to mitigate or halt harmful processes. These measures require a comprehensive knowledge of historical landscape changes, but also adequate management strategies. The required data was gathered from the sand grasslands of northern Serbia, as this habitat is of high conservation priority. This area also has a long history of different habitat management approaches (grazing and mowing versus unmanaged), which has been documented over of the last two decades. This dataset enabled us to quantify the effects of different measures across multiple taxa (plants, insect pollinators, and birds). We linked the gathered data on plants, pollinators, and birds with habitat management measures. Our results show that, at the taxon level, the adopted management strategies were beneficial for species richness, abundance, and composition, as the highest diversity of plant, insect pollinator, and bird species was found in managed areas. Thus, an innovative modelling approach was adopted in this work to identify and explain the effects of management practices on changes in habitat communities. The findings yielded can be used in the decision making as well as development of new management programmes. We thus posit that, when restoring and establishing particular communities, priority needs to be given to species with a broad ecological response. We recommend using the decision tree as a suitable machine learning model for this purpose.

## Introduction

As the natural vegetation of the forest-steppe biome in Central and Eastern Europe, Pannonian grasslands represent an important semi-natural habitat and biodiversity hotspot in the European agricultural landscape [1, 2]. Since the Anthropocene, as a result of human activities (e.g. grazing, mowing, burning) grasslands have expanded across Europe [3, 4]. However, land use changes that culminated in the intensification of agriculture in 20th century, triggered the

**Funding:** This work was supported by the Ministry of Science, Technological Development and Innovation of the Republic of Serbia [grant number 451-03-66/2024-03/ 200125 & 451-03-65/2024-03/200125]; Provincial Secretariat for Higher Education and Scientific Research,Autonomous Province of Vojvodina [grant number 142-451-3485/2023-01]. The funders had no role in study design, data collection and analysis, decision to publish, or preparation of the manuscript.

**Competing interests:** The authors have declared that no competing interests exist.

abandonment of grassland, afforestation, or conversion into arable land. Consequently, in the late 20th century, grassland ecosystems became habitats of high conservation value [5, 6].

Given the recent focus on biodiversity changes, grassland conservation offers a great opportunity to introduce innovative biodiversity management measures that can be implemented in the timeframe and conditions under which native species have evolved a mechanism to survive [7]. However, their design and implementation requires a comprehensive knowledge of the history of grasslands and their management, as well as its impact on the community composition of different taxa. For example, it has been established that preserved ancient grasslands are highly organised in terms of species composition and show a high degree of stability in relation to small-scale species fluctuations, while secondary grasslands are less resilient and their species pool is thus vulnerable to disturbances and immigration of other species [8–10]. Therefore, the species composition of these habitats is expected to respond differently to afforestation, and understanding this causality is crucial for the conservation and restoration of grassland biodiversity. Recent studies further highlight the importance of studying and monitoring different trophic groups, rather than basing management decisions on individual taxa [11–13]. Yet, despite the growing popularity and complexity of biodiversity research, most conservation efforts tend to neglect the role of different land uses in the biodiversity within these habitats. In recognition of this shortcoming, several authors have proposed development of conservation strategies for grassland working landscapes, i.e., ecosystems that are specifically managed with biological objectives [14, 15].

This proposal is justified, as grasslands play a vital role in providing a diverse range of ecosystem services essential for strengthening conservation efforts, actively involving local communities. These conservation initiatives encompass various ecosystem services, including biomass production and the provision of food for grazing animals and other herbivores, carbon storage and sequestration, as well as the creation of habitats for pollinators, migratory and breeding birds, among others [16]. They also highlight the importance of grasslands as a nesting and foraging habitat for solitary bees. Since numerous solitary bee species are currently facing threats, existing evidence suggests that generalist and highly mobile species may adapt to habitat loss by expanding their foraging ranges and utilizing alternative habitats and resources within the agricultural landscape [2, 17].

When developing these strategies, however, it is essential to recognise the role of controlled human activities—especially grazing and mowing [18–22] in maintaining the species richness and abundance of many taxa in grasslands [11, 23–25]. Still, extant management policies and conservation measures give precedence to the identification and analysis of rare or threatened species in protected areas, thus overlooking the fact that, in any process of natural habitat restoration through succession, it is the species with a broad ecological response that first conquer the habitat and form communities that create the micro-conditions for the specialists to enter. Consequently, their importance as precursors in natural succession processes is rarely examined in scientific studies.

This perspective adopted in both research and practice has created the need for decision makers to be actively involved from the beginning of the restoration process and to prioritise the habitat types to be restored and the species that should be returned to their original habitat. To exemplify how this strategy is adopted in practice, we have conducted our investigation in the Pannonian biogeographical region. This region has a complex, long, and documented history of landscape changes and thus provides a suitable site for biodiversity analyses guided by the following aims: (i) to investigate how management measures and habitat heterogeneity impact on grassland biodiversity; (ii) to evaluate the role of grazing and mowing in the restoration, development, and heterogeneity of semi-natural grasslands; and (iii) to assess the suitability of machine learning models for decision-making processes and management programs.

## Materials and methods

The data for testing the study hypotheses was gathered from the information sources on the state of nature in the Pannonian biogeographical region over a 300-year period. Plants, insect pollinators (wild bees and hoverflies), and birds were selected as model organisms, with the view that, while all taxa differ greatly in their resource use and mobility, they are equally important for the sustainability of grassland ecosystems [26].

### Study area

Subotica Sands which served as the study site is located in the far north of the Republic of Serbia, along the state border with Hungary (Fig 1). It is the southernmost part of sandy areas between the Danube and Tisa rivers [27] characterised by forest-steppes as the main native

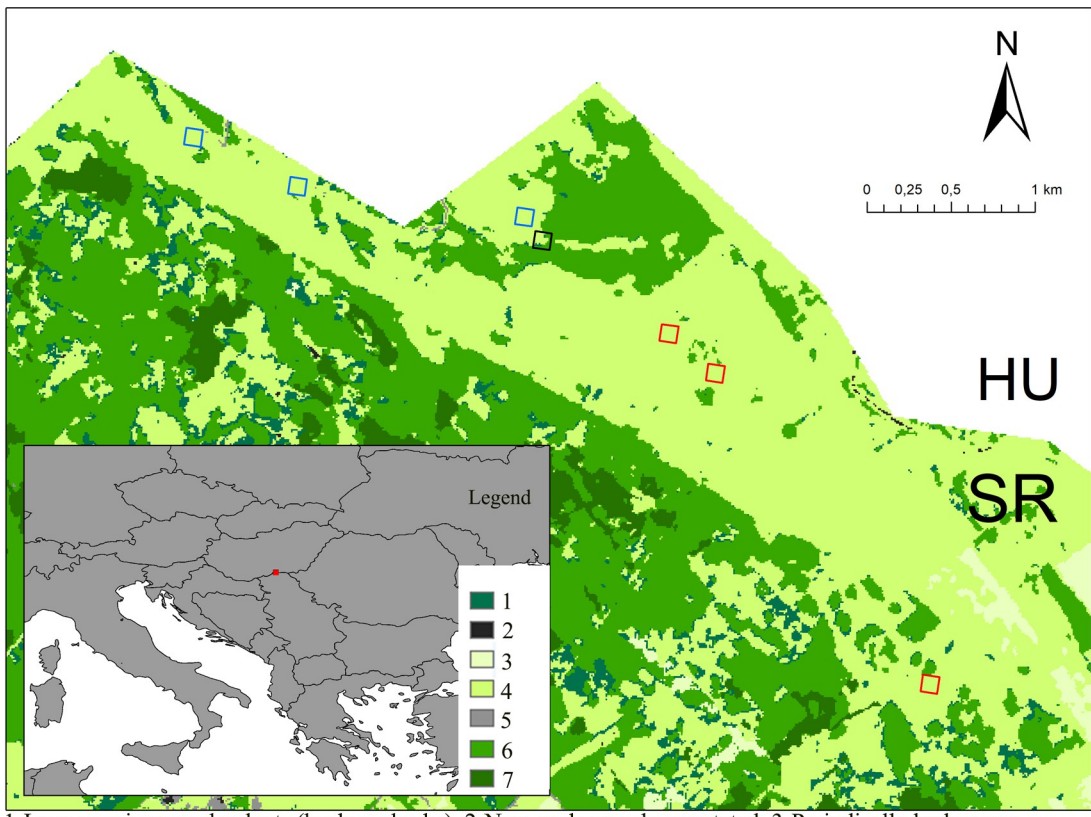

1-Low-growing woody plants (bushes, shrubs), 2-Non- and sparsely-vegetated, 3-Periodically herbaceous, 4-Permanent herbaceous, 5-Sealed, 6- Woody-Broadleaved deciduous trees, 7-Woody-needle leaved trees

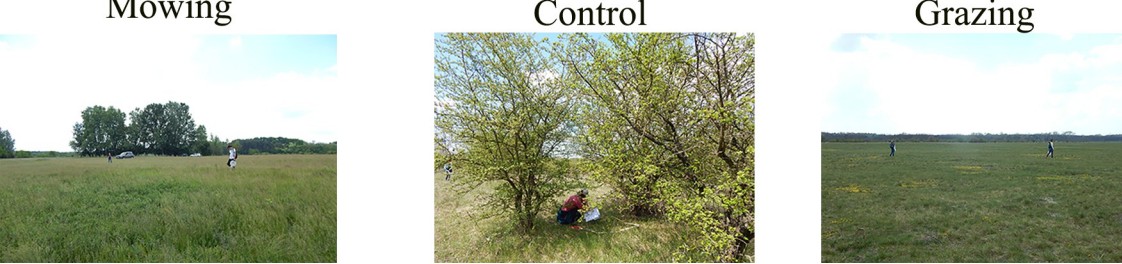

**Fig 1. Location of the study area and specific sites (blue square– mowing, black square – control, red square –grazing) within the Subotica Sands (Serbia).** Land-use data were obtained from Copernicus Land Monitoring Service (https://land.copernicus.eu/).

habitat type, with open grasslands—including sands, steppes, meadows, and marshes—intersected by woodlands and forests. As established by Butorac and Panjković [28], *Aceri tatarici-Quercion* and *Festucion rupicolae* mosaics have historically formed the climate-zonal vegetation of this area. However, *Convallario-Quercetum roboris* woods are currently present as remnants of former woodlands, while its secondary developed scrub (*Pruno spinosae-Crataegetum*) predominated [29]. Due to the vicinity of groundwater, continental saturation of soil with moisture, and specific relief with many micro-depressions, riparian forests and thickets, such as white poplar (ass. *Populetum albae*) or grey sallow (ass. *Salicetum cinereae*) stands, as well as marsh vegetation (ass. *Carici elatae-Fraxinetum angustifoliae*), remain in small fragments [29, 30].

Despite this evident decline in native habitats, it took almost two decades of work and harmonisation of conservation policies to finally have this area protected as a region of forest conservation value in 1982. Subsequent scientific research has shown that the area is home not only to forest species, but also to extremely small populations of species of international importance (spring meadow saffron *Colchicum bulbocodium* subsp. *Versicolor*, lesser blind mole-rat *Spalax leucodon*) that are restricted to small areas. In line with the latest trends in nature conservation and guided by the advances in scientific knowledge, part of the area that had previously been converted into orchards and vineyards was ploughed up in 2002. This provided an opportunity to revise the protection status and in 2003 the area was declared an Outstanding natural landscape. Active management measures were then developed and successively introduced. In the subsequent period, undergrowth was removed and large areas of Subotica Sands were mowed and levelled. Alongside these measures, in 2017, sheep and goat herds were allowed to actively graze in this area. In an event of a long winter, grazing was supplemented by regular mowing. Currently, this area serves as pasture for about 270 sheep per 230 ha, which are guided by shepherds and are permitted to roam freely. After two decades, the aforementioned active management measures were successful in the restoration of the populations of the most endangered species, thus creating the conditions for the reintroduction of some species that have previously thrived in this area (e.g., European ground squirrel *Spermophilus citellus*). Thus, this study site was not only suitable for the current investigation, it is also in line with the EU regulations, within which Pannonian sands and steppes are described as natural habitats with priority [28].

## Study design

For the present investigation, we selected seven sampling sites, each covering an area of one hectare, all situated within the Level I: Total/Strict protection regime (constituting 8.32% of the Subotica Sands area). However, these sites varied in terms of the management practices they had experienced in the past two decades. To provide more insight into the habitat characteristics, we categorized them into three groups: three sites were subject to mowing (M), three to grazing (G), and one served as a control site ©, having no history of specific management measures. During our fieldwork, we employed habitat classification to further specify the characteristics of each site according to National classification of Republic of Serbia (Table 1). Additionally, we established a buffer zone, extending 150 meters from the boundaries of each sampling site. This buffer zone provided a broader-scale perspective, allowing us to assess the impact of habitat features on bird and pollinator diversity. Furthermore, this approach allowed us to quantify parameters such as distance to the nearest trees, overall tree and shrub coverage, and the proportion of tree and shrub coverage within and outside of the study sites. The information required for these calculations was obtained by interpreting orthophoto images sourced from Google Earth Pro version 7.3.3.7786.

**Table 1. Habitat types at investigated sites in the study area.**

| Site | Habitat type *according to the National classification of Republic of Serbia (Official Gazette 2010/35)* |
|---|---|
| Grazing 1 | Panonian steppes on the sand* |
| Grazing 2 | |
| Grazing 3 | |
| Mowing 1 | |
| Mowing 2 | |
| Mowing 3 | |
| Control | Deciduous and xerophilous scrub of Hawthorn and Blackthorn |
| Buffer zone | Pannonian forest of Peduculate oak and Narrow-leaved ash |

*6260 NATURA 2000 habitat type

In the study area, two primary management practices were implemented. Mowing was conducted twice a year, which typically occurred in June and September. In addition, the grasslands experienced moderate grazing, with an average stocking density of 0.6 ewe/lamb pairs per hectare. These grazing activities were carried out under the supervision of shepherds, who ensured that the herds were led to designated pasture locations for specific periods of the year. At the control site C, no management measures have been carried out for 20 years, and the process of secondary succession has led to the formation of the forest-steppe. During 2021 in focus of this investigation, five field trips (in March, April, May, June, and September) were conducted.

For plants, eight quadratic plots (1×1 m) were designated at each site during the first field trip, resulting in 224 plots in total. At each plot, we recorded maximum vegetation height (MVH), total vegetation cover (TVC), and coverage percentage for each flowering species. Flowering plants were collected, labelled, and identified in laboratory by consulting relevant literature [30–37]. In addition, in order to gain insights into community dynamics and ecological characteristics, we employed Social Behavior Types (SBT) for all plant species. SBT categorizes plant species based on their roles in the community and provides valuable information about community richness, stability, naturalness, niche occupancy, and response to disturbances or deviations from the natural state [38]. All plant material was deposited at Herbarium BUNS, University of Novi Sad, Faculty of Sciences.

During all field visits, diversity and population abundance of pollinators (wild bees and hoverflies) were observed and samples were collected for 60 minutes per site, along two transects of 500 m length. Transect walks were conducted between 9:30 AM and 5:00 PM on sunny days (with over 30% sunlight) when wind speeds were low, and temperatures were above 13˚C if it was sunny and above 17˚C if it was cloudy. During the summer, transect walks and observations were conducted between 7:00 AM and 12:00 PM to avoid high temperatures that reduce bee activity levels [39]. All pollinators were identified to the species level in the field, while some of them were captured using a net and were transported to the laboratory in vials filled with ethyl acetate for identification to the species level. For the identification of solitary bees, we used the following resources: Mauss [40], Amiet [41], Schmid-Egger [42], Scheuchl [43] and for hoverflies, we referred to the following publications: Vujić [44], Speight and Sarthou [45], Likov et al. [46]. The captured specimens were subsequently deposited at the Department of Biology and Ecology, University of Novi Sad, Faculty of Sciences.

The diversity and abundance of birds breeding on the surveyed sites were assessed based on territory mapping. For this purpose, sites were visited on four occasions during the breeding season, between April 15th and June 15th. Visits were conducted at the peak of daily bird

activity (from 6 to 10 AM). Birds were mapped during a 5- to 15-minute walk around the site perimeters (100 × 100 m). All recorded (observed or heard) individuals, pairs, nests, or families with fledglings were precisely mapped using an orthophoto map of the landscape where individual trees, bushes, or other structural features were clearly visible. Only records within a 100 m buffer from site boundaries were included in the analysis. Atlas codes that describe breeding probability [47] were attributed to all records. Migrating species and species observed only overflying were excluded from the analysis.

## Data analysis

Species diversity were calculated as Shannon-Weaver index and was analysed for all study sites, while differences across sites and between management measures were tested through one-way ANOVA using Duncan's test (P ≤ 0.05).

To assess the species similarity among sites grouped by management measures, Jaccard similarity index was adopted, whereby a value in the 0−0.25 range indicates small similarity of species composition between sets, 0.25−0.5 is interpreted as medium dissimilarity, 0.5−0.75 indicates medium similarity, and 0.75−1 is indicative of high similarity. In other words, values close to zero are associated with heterogeneous structures, while those close to unity imply homogenisation.

To investigate the relationship between species (pollinator and bird) diversity and habitat variables (including plant diversity), we employed generalised linear models (GLMs), based on the Gaussian distribution family. Separate models were constructed for the responses of pollinators and birds, and the best model for each response variable was selected based on Akaike's information criterion corrected for small sample size (AICc) [48]. The AICc considers both the goodness of fit of the model and the number of parameters used, with lower values indicating a better trade-off between model complexity and fit to the data.

Linear discriminant analysis (LDA) was performed to quantify and describe sites under different management measures, with maximum vegetation height, total vegetation cover, and Shannon-Weaver index treated as predictor variables. Before performing the LDA analysis, data were pre-processed by scaling all the predictor variables. The dataset was then partitioned into a training set (consisting of 75% of all available observations) and a testing set (comprising the remaining 25%). The training set was used to fit the LDA model, while the testing set was used to evaluate the model performance. A ten-fold cross-validation method was adopted during the training process to tune the model hyperparameters and prevent overfitting. Finally, the accuracy of the LDA model was measured by computing the proportion of correctly classified observations in the testing set.

Statistical analyses (ANOVA, GLM, and LDA) were performed in R version 4.2.2 [49] using dedicated packages "vegan" [50], "agricolae" [51], "dplyr" [52], "ggplot2" [53] and "MASS" [54].

A decision tree for plants (as habitat builders) was developed in order to analyse species frequency and their causal relationship at three levels: individual plots, site level, a habitat management measure (G, M, or C). This method aimed to pinpoint vital plant species linked with various habitat management approaches. Decision trees are popular in ecological modeling, providing a visual representation of decision-making processes, particularly valuable for researchers and conservationists. These hierarchical structures involve attribute tests, branching for outcomes, and leaf nodes for predictions [55]. In ecology, they aid in tasks like species classification, habitat assessments, and biodiversity analyses, known for their interpretability even for non-experts. This interpretability is vital for informed conservation and management decisions, allowing stakeholders to comprehend strategies. By identifying key species, habitat

preferences, and ecological connections, decision trees facilitate ecosystem management, bridging the gap between research and practical conservation efforts, ultimately supporting biodiversity preservation and ecological sustainability. In the field of computer science, decision trees are artificial intelligence models which belong to the machine learning subbranch of the artificial intelligence field. They are not hand-made, but rather trained on our data in order to give us meaningful insights and predictions. From the inputs and outputs of setup decision tree model, thresholds for decision making inside the model are automatically learned to minimize error which in our case is classification loss.

For this analysis, as described later, we used Python programming language. As the dataset used in this study exhibits class imbalance, several methods were explored to overcome this issue. The imbalance comes from the nature of the study, with one control set of observations compared to three for the other output classes (mowing, grazing). In the process of training our Decision Tree model, we tested various approaches to overcome the class imbalance of the data. Initially, we attempted using the Decision Tree classifier's inherent feature to apply class weights, assigning more weight to the minority class. This was achieved by controlling the 'class_weight' parameter in the 'scikit-learn' library implementation of the Decision Tree Classifier. In the model setup, we assigned three times the weight to the Control output class compared to the other two classes. Unfortunately, this adjustment did not lead to improved performance. To overcome this problem, we explored both undersampling and oversampling techniques. Undersampling posed a risk of losing crucial information necessary for effective model learning due to the dataset's limited size. To address the class imbalance, we employed the SMOTE [56] oversampling technique from the 'imblearn' Python library. Given the restricted variety in the control data, obtaining more diverse and substantial data in the future would be advantageous in alleviating the class imbalance issue. We adopted the decision tree implementation from the "scikit-learn" [57] package and relied on the "pandas" [58] and "numpy" [59] libraries for data preprocessing. In addition, for exploratory data analysis in preparation for model training, "ydata_profiling" package [60] was adopted. Due to the tabular nature of our dataset, we opted for decision tree-based algorithms as they provide good balance between performance and interpretability. These models were used as classifiers, whereby the training phase allows the utility of certain species for segregation of different habitats to be ascertained. During the learning process, as will be described in the sections that follow, we monitored model accuracy, as well as assessed the importance of different features and their gini values for identifying the species that are most relevant for differentiating habitat types.

## Results

We observed variations in biodiversity depending on the management measures adopted at a particular site. In total, we recorded presence of 164 vascular plant species, 31 insect pollinator species, and 61 bird species (S1 Table). At 85, 14, and 26 (G), and 115, 21, and 19 (M), the greatest number of plant, pollinator, and bird species was found at the grazed and mowed sites, respectively, while at unmanaged site (C), only 59, 9, and 18 species were recorded.

The greatest differences between the sites were found in the plant species, namely the shrubs such as *Cornus mas*, *Crataegus monogyna*, *Rosa canina*, *Prunus spinosa*, and *Fraxinus ornus*, which dominated within the unmanaged area, while in the grazed and mown areas they appeared only sporadically and mainly as young plants. At all sites (especially those under grazing), prevalence of species from the Poaceae family was noted, and *Festuca* was the predominant plant genus. Maximum vegetation height (MVH) ranged from 12 to 80 cm, while total vegetation cover (TVC) ranged from 10% to 100%. The G sites exhibited the significantly

lowest values for both MVH and TVC, whereas the C site recorded the significantly highest values (Fig 2).

Most of the plants found at all sites belonged to disturbance-tolerant and generalist SBTs, while specialist species and IAS were represented with a small share, approx. 7% and 5%, respectively (Table 2). Greater percentages of natural pioneers were found at managed compared to unmanaged sites.

The recorded 32 pollinator species (20 wild bees and 12 hoverflies) consisted of 290 individuals (269 wild bees and 21 hoverflies). The most common wild bee families were Apidae and Megachilidae, while only one species was detected from the family Halictidae. Most of the species had a very low abundance, with 25 species having 10 or fewer individuals, five species having 10 to 30 individuals, and only one species having over 100 individuals.

In total, 46 species of birds were possibly breeding on the surveyed sites, 31 of which were confirmed or probable breeders. Passerines were the dominant group with 33 species. Bird assemblage was dominated by species typically inhabiting forests, woodlands, or shrubs (28

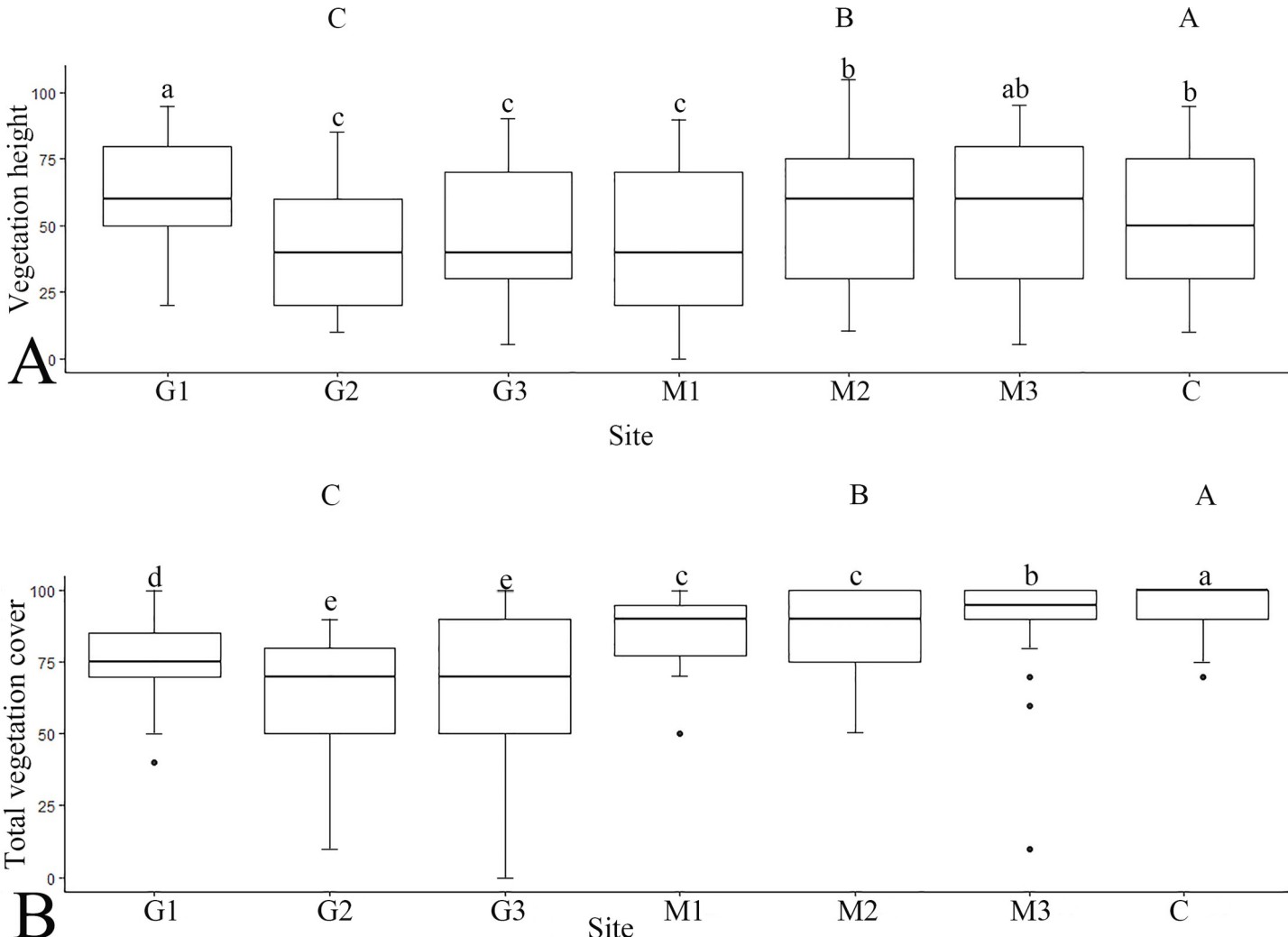

**Fig 2. Vegetation height and total vegetation cover on the study sites (M – mowing, C – control, G – grazing).** *Different lowercase letters indicate significant differences between sites (one-way ANOVA; p < 0.05); Different uppercase letters indicate significant differences between management practices (one-way ANOVA; p < 0.05).

**Table 2. Relative cover (%) of each social behaviour type (SBT) for plants found within management practices (G–grazed, M–mowed, and C–control).** S–specialists; G–generalists; DT–disturbance-tolerant species; NP–natural pioneers; IAS–invasive alien species.

| SBT | G | M | C |
|---|---|---|---|
| S | 7.06 | 7.82 | 3.39 |
| G | 14.12 | 25.22 | 27.12 |
| DT | 25.88 | 24.35 | 32.20 |
| NP | 12.94 | 10.43 | 8.47 |
| IAS | 5.88 | 5.22 | 1.69 |

species), while grassland birds were significantly less abundant (six species). Twelve species were present at most study sites, while 13 species were recorded at only one site.

The Shannon-Weaver index for plants was significantly higher at mowed sites than at grazed and unmanaged sites (Table 3). Its values also varied among sites, whereby the lowest value was recorded at the G1 site and the C site (0.96 and 0.97, respectively), while the highest value was associated with the G3 site (1.28). In general, no significant differences in the Shannon-Weaver index for plants were noted between mowed and grazed sites, or for pollinators and birds across any sites, suggesting that there are no consistent differences in diversity within taxa that could potentially be attributed to the adopted management measures.

Our investigation based on the total species diversity findings and the analysis of recorded species by site according to management type revealed that the community similarity among G, M, and C sites is small. Among the investigated groups, birds were found to be the least heterogeneous (with the highest index of community similarity, i.e., above 0.5), while for plants and pollinators this index ranged from 0.25 to 0.5 (Table 4). When similarities at the group level were examined, a clear pattern was established for each group. Specifically, the greatest similarity among plants was noted between G and M, while pollinators were most similar at G and C sites, and M and C emerged as most similar in terms of bird populations. Plant analyses further revealed that at the G and M sites grassland was the predominant floristic structure, while the C site was characterised by forest degradation. According to the findings related to pollinators, community similarity index is uniform across the entire study area.

According to the GLMs, no significant trends can be established for pollinator diversity and habitat parameters. On the other hand, GLM findings related to bird species revealed that their diversity was primarily affected by diversity of plants within the sites and the percentage of trees outside the site boundaries (Table 5).

**Table 3. Mean values of the Shannon-Weaver index for plants, pollinators, and birds at study sites (G–grazed, M–mowed, and C–control).**

| Site | Plants | | Pollinators | | Birds | |
|---|---|---|---|---|---|---|
| G1 | 0.96 b | | 0.94 a | | 0.85 a | A |
| G2 | 1.09 ab | AB | 0.44 a | A | 1.91 a | |
| G3 | 1.28 a | | 0.93 a | | 1.76 a | |
| M1 | 1.19 ab | | 0.92 a | A | 1.75 a | |
| M2 | 1.06 ab | A | 0.52 a | | 1.68 a | A |
| M3 | 1.30 a | | 1.36 a | | 1.65 a | |
| C | 0.97 b | B | 0.70 a | A | 2.00 a | A |

* Different lowercase letters indicate significant differences between sites (one-way ANOVA; p < 0.05); Different uppercase letters indicate significant differences between management practices (one-way ANOVA; p < 0.05)

**Table 4. Community similarity for plants, pollinators, and birds at study sites (G–grazed, M–mowed, and C–control).**

|  | Plants | | Pollinators | | Birds | |
|---|---|---|---|---|---|---|
|  | G | C | G | C | G | C |
| C | 0.26 | | 0.29 | | 0.46 | |
| M | 0.36 | 0.28 | 0.24 | 0.24 | 0.62 | 0.64 |

The LDA results facilitated a separation of the three habitat management types. The first axis of the discriminant analysis explained over 95% of the total dispersion, while the combination of the first two axes accounted for 100% of the variability (Table 6). All variables were more closely related to sites managed through mowing. While a small overlap was detected between grazed and mowed sites, it was much broader between mowed and unmanaged sites (Fig 3). It is also worth noting that the unmanaged area was relatively uniform compared to others, as much greater variability can be observed in the results related to both M and G sites. Among the predictor variables, total vegetation cover and diversity of plant species had the highest factor scores on the first axis, and according to this decision tree model was trained only on data related to plants.

The decision tree model was trained in a classification setting where the habitat management measure labels were used as the algorithm output and the remaining dataset was used as the input. By conducting post-training analysis, we were able to extract and rank features by their relative importance, allowing us to determine which plant species are most relevant for different habitats. Although several model were trained and used during our experiments—including Random Forest and XGBoost [61] Gradient Boosted Tree models—Decision Tree Classifier model with maximum tree depth set to 30 proved most useful. While the two ensemble models provided a slight increase in accuracy, we opted for the Decision Tree Classifier as it is fully interpretable and provided good validation accuracy (71%). When applying this model, the data was split into training and validation sets at a standard 80/20 ratio. For all model hyperparameters, a combination of manual tuning and grid search was used to find the most optimal values. Data was not normalised as all tested models performed well when applied to raw data.

**Table 5. Results yielded by GLM analysis of the effect of habitat characteristics on the pollinator and bird species diversity within the study area.**

|  | Estimated parameter | Standard error | t value | p value |
|---|---|---|---|---|
| **Diversity of pollinators** | | | | |
| Intercept | 5.52 | 1.33 | 4.16 | 0.000328*** |
| Distance to the nearest tree (m) | -0.0038 | 0.0019 | -2.01 | 0.055 |
| Trees outside sites | -0.043 | 0.0259 | -1.69 | 0.101 |
| **Diversity of birds** | | | | |
| Intercept | -0.72 | 0.66 | -1.08 | 0.29 |
| Trees outside sites (%) | 0.02 | 0.009 | 2.28 | 0.03* |
| Vegetation height | -0.01 | 0.0076 | -1.67 | 0.11 |
| Plant diversity | 2.10 | 0.50 | 4.15 | 0.00035*** |

Significance is expressed as

* p < 0.05,

** p < 0.01, and

*** p < 0.001.

**Table 6. LDA results with proportion of dispersion and variable scores on the first two axes.**

|  | LD1 | LD2 |
| --- | --- | --- |
| Proportion of total dispersion | 97.18 | 2.82 |
| Maximum vegetation height (VG) | -0.2862 | -0.4852 |
| Total vegetation cover (TVC) | -2.6927 | 0.3609 |
| Diversity of plants (DPl) | 0.4876 | 1.2224 |
| Diversity of pollinators (DPo) | 0.0495 | 0.1797 |
| Diversity of birds (DB) | -0.1617 | -0.5275 |

According to the obtained findings, *Galium verum* was the most frequent plant species, as it occurred in 179 of the 224 sample units. The dichotomous branching emanating from this species clearly shows the separation of the G site in relation to the M and C sites (Fig 4). Depending on whether it occurs in a community with the species *Dactylis glomerata* or *Eryngium campestre*, the branch is recognised as G or M, respectively. Following *D. glomerata*, G is characterised by the presence of *Veronica arvensis, Taraxacum officinale, Phragmites communis, Rumex acetosa, Crataegus monogyna, Bromus ramossus, Carex caryophyllea, V. verna*, and *V. vindobonensis*. On the other hand, following *Eryngium campestre*, M is characterised by *Falcaria vulgaris, Rubus* sp., *Fraxinus ornus, Knautia drymeia, Teucrium chamaedrys, Thalictrum elata*, and *Calamagrostis epigeios*. Counter to the bilinear distinction of G and M, C is not recognised as a single strong branch, but rather consists of semi-branches that have emerged

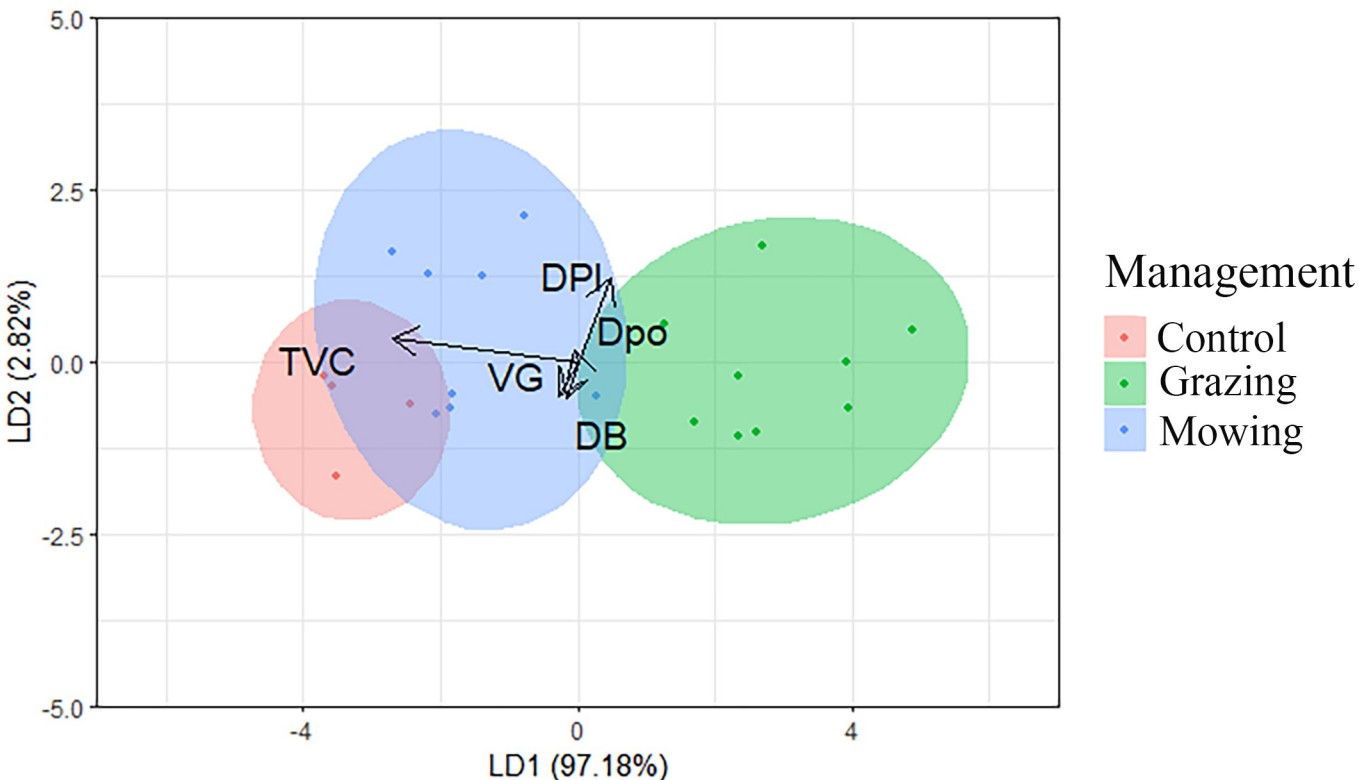

**Fig 3. Plots of the first two discriminant functions (LD1, LD2) yielded by Linear Discriminant Analysis (LDA).** M – mowing, C – control, G – grazing. TVC – Total vegetation cover; DPl – Diversity of plants; VG – Maximum vegetation height; DPo – Diversity of pollinators; DB – Diversity of birds.

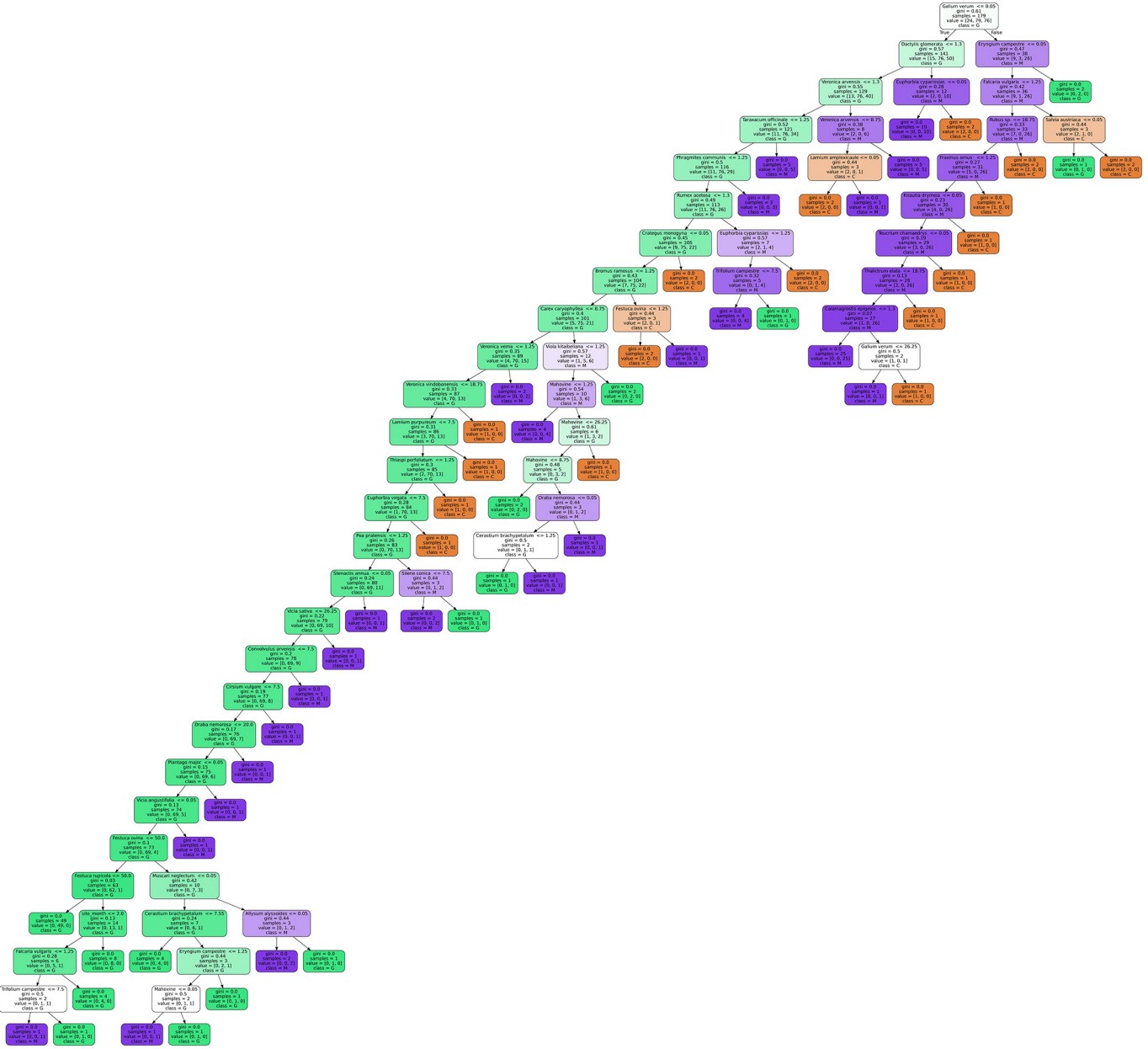

**Fig 4. Visualisation of the decision tree model for plant species observed and collected in the Subotica Sands study area.** Green, purple, and orange colour correspond to grazed, mowed, and control site, respectively. Colour intensity indicates stronger support of the selected branch (management measure) by the included plant species.

mainly from M branches. It is defined by three herbaceous species: *Lamium amplexicaule*, *Salvia austriaca*, and *Festuca ovina*.

## Discussion

Understanding the relationship between biodiversity and management practices can provide valuable insights for improving monitoring strategies, as well as for developing early warning systems and management strategies for *In-situ* conservation [62–65].

In our study area, the management measures that have been adopted in the last two decades (mowing and grazing) proved to be most significant for plants comparing to pollinators and birds. This is not surprising, given that in grasslands, grazing and trampling prevent secondary succession and the formation of monodominant shrub stands to the detriment of other steppe species [66]. In our study area, small trees and shrubs (*Cornus mas*, *Crataegus monogyna*, *Rosa canina*, and *Prunus spinosa*) were only recorded as individual juvenile plants. However, as the presence of shrub species is beneficial for pollinators and birds, individual specimens should be preserved in managed areas, as they are not only an important food source for pollinators in spring, but are also a natural component of these habitats. Similar conclusions were reached by other authors based on their experimental studies [67–69]. Available evidence also indicates that cattle grazing creates and maintains an open soil in the sandy grassland, which is an ideal substrate for the formation of wild bee nests, as it warms up quickly in spring and retains heat for a long time. Within the area in focus of the present study, with the shrub species and the plant species that flower in late winter and early spring (*Colchicum bulbocodium* subsp. *versicolor*, *Gagea* spp., *Muscari neglectum*, *Veronica arvensis*, etc.), two conditions for the presence of wild bees are already fulfilled: a suitable place to nest and a food source. Therefore, we can expect a greater number of species (including specialists) to occur within the managed areas over time, especially on the grazed sites.

We also noted that the adopted management measures affected the occurrence of species characteristic of the habitat, especially the plant species. At the grazing sites, we found most of the plant species from the Poaceae family, the genera *Festuca*, *Bromus* and *Poa* in particular. These results may indicate that sheep favour herbs, due to which grazing is beneficial for the proliferation of grasses [70]. This assertion, if true, could explain the negative impact of grazing on nectar-dependent pollinators, even if it may facilitate restoration and maintenance of natural habitat types. According to Hellström et al. [71], sheep grazing may be an efficient strategy in the initial stages of restoration, since these animals preferentially feed on leaves and young saplings of trees and shrubs.

Our analyses focusing on insect pollinators showed a similar pattern of presence in relation to management measures and habitat type. Contrary to expectations, the abundance of pollinators, especially hoverflies, was low at all sites. This can be explained primarily by the dominance of Poaceae species in the study area, which are not important either as a food source or as a host for this group. Insect pollinator diversity is strongly linked to the availability and amount of floral resources which are in their foraging range, as well as the composition of floral communities, the shape and size of the fragment, and the availability of nesting sites. In areas with a low share of semi-natural habitats, pollinator abundance in isolated fragments is higher, which indicates that the source of pollen is limited in simpler areas [72]. As previously noted, habitats with open ground surfaces, such as those found at the studied sites, are optimal for wild bees. The presence of plants from the Asteraceae and Fabaceae families is also considered important for pollinators, as their flowers represent a high-quality resource. Indeed, this has been demonstrated at the mowed sites included in our investigation, where the highest numbers of plant species from these families as well as the highest number of pollinators were recorded. However, some authors have found that having cattle or horses as the main grazers yields better results in terms of plant and pollinator diversity, suggesting that this may be a viable habitat management strategy [73]. Moreover, the pollinator species that were registered in the study area are very important for the pollination of agricultural crops located in its vicinity [74].

Compared to plants and pollinators, birds use habitats on a larger spatial scale. We recorded presence of bird communities that are typical of semi-open landscapes dominated by forests and shrubs (forest and shrub generalists). Although the grassland patches may not have been sufficient for establishing a grassland bird community, certain species, such as *Coracias*

*garrulus*, *Lanius collurio*, and *Saxicola torquata*, were identified as potentially benefiting from grassland management due to their foraging habits. Our study highlights the significance of the Subotica Sands landscape for *Coracias garrulous*, a species that faced population decline in the past [75]. The mosaic landscapes of Subotica Sands remain among the few sites where this secondary cavity-nesting bird is not completely dependent on artificial nest boxes. Grassland management provides a unique opportunity for focusing on some species of conservation concern, even though management is not the main factor determining bird communities at the grassland level. The presence of species like *Alauda arvensis*, *Anthus campestris*, and *Coturnix coturnix* exclusively in grazed areas suggests that grazing may be more effective than mowing in restoring suitable vegetation structure, emphasizing the importance of adopting grazing as the primary management measure for habitat restoration. As birds are the most sensitive to changes in grassland management strategies [11], the presence of unique and more specialised species thus confirmed the importance of grazing, suggesting that it should be adopted as the main management measure for the restoration of mosaic habitat.

Although habitat heterogeneity is a mirror of the ancient natural vegetation, today it is a consequence of the long-term agricultural landscape changes. Thus, it is useful for describing spatial and temporal differences for different taxonomic groups from a historical perspective. Our analyses confirmed that some taxonomic groups (plants and pollinators) exhibit similar species structure patterns in differently managed habitats (based on a relatively low community similarity value), concurring with the previously reported findings [76]. These observations can be ascribed to more rapid changes in unmanaged grasslands, followed by general deterioration of this habitat. As birds are highly sensitive to changes in grasslands, these findings were expected and can also be explained by the landscape changes that began long ago, given that most of the semi-natural grassland areas have been fragmented in the last 50 years [29, 77, 78]. The results yielded by our LDA model further indicate a gradual change in the species structure in Subotica Sands, from unmanaged sites, through mowing, to grazed areas. This model also allowed us to determine that grazed areas are noticeably separated from mown and unmanaged areas. Similar patterns were previously confirmed in other study areas [12, 79–81], indicating that good habitat management is beneficial for biodiversity, and has the capacity to influence species richness, abundance, and community composition.

Our analyses further revealed that areas that have been under constant human pressure for a long time can be relatively quickly restored with appropriate protection measures. We noted many beneficial effects of such practices, including dominant presence of generalists, disturbance-tolerant plants, and natural pioneers, but also significant occurrence of specialists. The presence of first three SBTs in Subotica Sands in the timeframe of 20 years is attributed to the ploughing performed in 2002, followed by continued grazing and mowing. As pointed out by Kapás et al. [82], grazing supports the establishment of grassland species on restored sites, which will subsequently serve as dispersal vectors for seeds. Therefore, they will assist plant species establishment by generating disturbances on the surface, and will restrict and/or favour other plant species, as confirmed in our research. Our findings also concur with the view put forward by Littlewood et al. [83], who argued that, without the help of grazers, it is difficult for specialists to gain space in the dense vegetation dominated by non-specialist grassland species. The small number of specialists and a high percentage of disturbance-tolerant species in unmanaged site compared to managed sites also points towards a general deterioration of grassland species composition in abandoned grasslands and possibly a lack of remnant populations [84–86]. Although specialists are important habitat edificators, extant research suggests that habitat restoration initiatives must primarily consider species with a broad ecological response.

While these outcomes can be extracted from the existing biodiversity databases, as they are usually large repositories of different data types, including different taxa and investigated

territory, they are difficult to analyse without extensive support from latest technological tools. With the advent of artificial intelligence and machine learning, these drawbacks have been mitigated, and in this study, we adopted the Decision Tree Machine Learning Model, as we hope that such innovative approaches will soon become the norm in the ecological restoration process and will be used by decision makers in the design of different management strategies. This particular model allows researchers to visualise scientific data and propose explainable relationships between species, which could be further incorporated into management policies. From a conservation perspective, this model was particularly useful in establishing that generalist and disturbance-tolerant species are habitat builders, thus confirming their vital role in the restoration of a particular habitat type. Furthermore, this analysis has shown that, although different species occur at different sites, they all have the same ecological preferences, and are well known and easily recognised in the field. Since woody species were found mostly within unmanaged sites (which in this study correspond to forest-steppes) and specialists were recorded only at individual locations (mainly within managed sites, i.e., restored open grasslands), the decision tree model was appropriate for the analyses, as it allowed us to focus on herbaceous and shrubby species only.

## Conclusions

The present study has confirmed that clear differentiation of grazing areas (in terms of structure and number of species) remains the most important management measure, as it leads to the formation of the desired habitat types (open grass habitats of the Pannonian biogeographical region). We also conclude that the cattle used for habitat management today are an adequate substitute for the former cattle species (*Bos primigenius*, *Bos taurus*) that colonised the Pannonian Plain together with humans in the Anthropocene. Therefore, modern management measures should follow this practice, i.e., depending on the type of habitat and the desired outcome, focus should be given to measures that are a modern substitute for former natural activities if these are not feasible.

## Supporting information

**S1 Table. List of recorded species at Subotica sand (Serbia).**
(PDF)

## Acknowledgments

The authors gratefully acknowledge the support of Sandra Čokić Reh, Tamas Vinko, Otto Szekeres, Ivan Laslo Pajić, and Saša Vujić from Outstanding natural landscape Subotica sands during fieldtrips. We also wish to thank Zsolt Molnár from Centre for Ecological Research, Institute of Ecology and Botany, Vácrátót, Hungary, for his assistance with botany research, and express our gratitude to Klára Szabados from Institute for Nature Conservation of Vojvodina province for her endorsement of this research project.

## Author Contributions

**Conceptualization:** Dubravka Milić, Milica Rat, Dimitrije Radišić.

**Data curation:** Dubravka Milić, Milica Rat, Bojana Bokić, Sonja Mudri-Stojnić, Dušan Jakovetić, Boris Radak, Tamara Tot, Dimitrije Radišić.

**Formal analysis:** Dubravka Milić.

**Investigation:** Dubravka Milić, Milica Rat, Bojana Bokić, Sonja Mudri-Stojnić, Boris Radak, Tamara Tot, Dušanka Vujanović, Goran Anačkov, Dimitrije Radišić.

**Methodology:** Dubravka Milić, Milica Rat, Dimitrije Radišić.

**Software:** Nemanja Milošević, Nataša Sukur, Dušan Jakovetić.

**Supervision:** Dubravka Milić.

**Visualization:** Dubravka Milić, Nemanja Milošević, Nataša Sukur, Dušan Jakovetić.

**Writing – original draft:** Dubravka Milić, Milica Rat, Bojana Bokić, Sonja Mudri-Stojnić, Nemanja Milošević, Dimitrije Radišić.

**Writing – review & editing:** Dubravka Milić, Milica Rat.

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
