## [Decision Letter · Decision Letter 0]

19 Oct 2023

PONE-D-23-26216Exploring the effects of habitat management on grassland biodiversity: A case study from northern SerbiaPLOS ONE

Dear Dr. Milić,

Thank you for submitting your manuscript to PLOS ONE. After careful consideration, we feel that it has merit but does not fully meet PLOS ONE’s publication criteria as it currently stands. Therefore, we invite you to submit a revised version of the manuscript that addresses the points raised during the review process.

We look forward to receiving your revised manuscript.

Kind regards,

Tzen-Yuh Chiang

Academic Editor

PLOS ONE

Journal Requirements:

"This work was supported by the Ministry of Science, Technological Development and Innovation of the Republic of Serbia [grant number 451-03-47/2023-01/200125]; Provincial Secretariat for Higher Education and Scientific Research,Autonomous Province of Vojvodina [grant number 142-451-3161/2022-01]"

6. We note that [Figure 1] in your submission contain [map/satellite] images which may be copyrighted. All PLOS content is published under the Creative Commons Attribution License (CC BY 4.0), which means that the manuscript, images, and Supporting Information files will be freely available online, and any third party is permitted to access, download, copy, distribute, and use these materials in any way, even commercially, with proper attribution. For these reasons, we cannot publish previously copyrighted maps or satellite images created using proprietary data, such as Google software (Google Maps, Street View, and Earth). For more information, see our copyright guidelines: http://journals.plos.org/plosone/s/licenses-and-copyright.

Reviewers' comments:

Reviewer's Responses to Questions

**Comments to the Author**

1. Is the manuscript technically sound, and do the data support the conclusions?

Reviewer #1: Yes

Reviewer #2: Partly

2. Has the statistical analysis been performed appropriately and rigorously? 

Reviewer #1: I Don't Know

Reviewer #2: Yes

3. Have the authors made all data underlying the findings in their manuscript fully available?

Reviewer #1: Yes

Reviewer #2: Yes

4. Is the manuscript presented in an intelligible fashion and written in standard English?

Reviewer #1: Yes

Reviewer #2: Yes

5. Review Comments to the Author

Reviewer #1: This study focuses on the effects of different grassland management types, specifically grazing and mowing compared to control, on the diversity of three different groups: plants, pollinators, and birds. They show differences in the diversity of these three groups in response to management. In my specific comments on the paper I've highlighted places where I think additional methodological details are needed.

Reviewer #2: Exploring the effects of habitat management on grassland biodiversity: A case study from northern Serbia

This an interesting study presenting the effect of grassland management (mowing and grazing) on different biodiversity taxa. Although the study is valuable, it could be much improved. Some parts of the manuscript are not easy to follow, especially the introduction, due to poor writing. And the discussion section is too lengthy, should be shortened and more focused to the main questions assessed in the study. But the main concern is the sampling design, which is unbalanced (only 1 control site vs. 3 grazing and 3 mowing sites) and this should be acknowledged and dealt with proper statistical analysis in the study.

Minor comments

L81: I recommend changing the sentence to make it clearer. By “initiatives” you refer to the conservation strategies? Something like “these conservation initiatives cover ecosystem services such as biomass production and…”, as you are referring here to the ecosystem services grasslands provide

L84 To what species you are referring to by “as many species are presently threatened”? Of solitary bees?

L83-87 the whole sentence would benefit from some rephrasing

L86. Fix typo, forging should be “foraging”

L167: fix typo, nearest tree (NOT three)

L171-175: treatments could be explained in a more structured and clear way, without mixing information

L186-192: no information on time of the day when pollinator sampling was conducted, please add

L205: I suggest rephrasing, species diversity was "calculated" instead of interpreted.

L249: In the study area is not needed, I suggest to delete that

L250. Please rephrase. Recorded instead of “noted presence”

L253: Please correct. Unmanaged site, singular as there was only one site.

Table 1. Are those numbers mean % across sites? Please specify. Also, would be better if the description of social behaviour types in table legend text follows the order of appearance of the table, starting with S, D, DT, NP and IAS

Table 4. Fix typo, distance to nearest TREE

Fig. 3. IT would help to add the full name of management type in the legend in the figure: Control, Grazing, and Mowing. Also Variat should be replaced by Management.

Discussion could be shortened in order to improve flow and focus on main messages.

6. PLOS authors have the option to publish the peer review history of their article (what does this mean?). If published, this will include your full peer review and any attached files.

Reviewer #1: No

Reviewer #2: No

---

## [Author Response · Author response to Decision Letter 0]

15 Dec 2023

Dear Editor,

I have received the reports of you and two reviewers on the manuscript “Exploring the effects of habitat management on grassland biodiversity: A case study from northern Serbia”. I am glad to see that manuscript could be reconsidered for publication if revisions were prepared. Following reviewer’s comments strictly, we made certain changes in the manuscript. 

Land-use data for Figure 1. were obtained from Copernicus Global Land Service (https://land.copernicus.eu/). According to 4.1 Terms of Use of Copernicus Global Land Service: The product(s) described in this document is/are created in the frame of the Copernicus programme of the European Union by the European Environment Agency (product custodian) and is/are owned by the European Union. The product(s) can be used following Copernicus full free and open data policy, which allows the use of the product(s) also for any commercial purpose. Derived products created by end users from the product(s) described in this document are owned by the end users, who have all intellectual rights to the derived products (https://land.copernicus.eu/en/technical-library/product-user-manual-for-clc-backbone-raster-only/@@download/file). The sources of the layers used for map creation, including cities, are specified in the caption of Figure 1. Furthermore, the map has not been previously copyrighted to our knowledge. 

Together with the revised manuscript, we are sending a list of responses to the comments. We have endeavored to respond to all the points raised. As detailed below, we have checked all the general and specific comments provided by the Referees and have made the necessary changes according to their recommendations. 

The authors would like to express their appreciation to the reviewers and editors for their help and useful comments.

Sincerely,

Dubravka Milić

Reviewers' comments:

Reviewer #1: This study focuses on the effects of different grassland management types, specifically grazing and mowing compared to control, on the diversity of three different groups: plants, pollinators, and birds. They show differences in the diversity of these three groups in response to management. In my specific comments on the paper I've highlighted places where I think additional methodological details are needed.

Line-by-line comments:

Reviewer comment: Line 86: change “forging” to “foraging”

Milić et al.: corrected. We accept the Reviewer's remark.

Reviewer comment: Line 124: I’m not sure what is meant by “degradation stadium”

Milić et al.: Pruno spinosae-Crataegetum is degradation stadium of Convallario-Quercetum roboris. We rephrased the sentence to provide a clearer explanation.

Reviewer comment: Figure 1: It’s difficult to see the labels in the figure that show where the 3 different treatments are located. Can the figure be edited to make these more readable?

Milić et al.: We modified Figure 1 to align with the journal's requirements and enhance its readability. 

Reviewer comment: Line 167: The “nearest three” what? Nearest three other sites?

Milić et al.: It was our mistake, we should have written “the nearest trees”. We have corrected in our manuscript.

Reviewer comment: Lines 162-170: I would like to see more detail here because I don’t think these methods are clear as currently written. How were the habitats characterized? It sounds like there is some focal habitat that was characterized, as well as a 150 m buffer zone that presumably was also characterized by habitat type. Did the measures of tree/shrub cover determine the characterization of habitat types? How large is the focal study area at each of the sites?

Milić et al.: We accept the Reviewer's remark. Throughout the 'Study Area' section, we have made modifications to the text and added an additional table. In this table, we have categorized the habitat types according to our National classification of Republic of Serbia and mark which habitat type corresponds to NATURA 2000 habitat type. This was done to facilitate the identification of habitat types within the research area. Furthermore, we wrote that each sampling site covering an area of 1 hectare. 

Reviewer comment: Line 183: I’m not familiar with “social behavior types” for plants. What is this?

Milić et al.: Social behavior types (SBT) are defined by the role that a plant species plays within its community. These categories represent how a particular plant is interconnected with its habitat, conveying valuable information about the naturalness of this connection. The presence of these SBTs within a community can provide insights into various aspects, such as the ecological richness of the community, its stability, its natural state, niche occupancy, regenerative potential, and the degree of disturbance, transformation, or deviation from the natural state. According to Borhidi (1995), these categories include specialists (S), generalists (G), disturbance tolerants (DT), natural pioneers (NP), natural competitors (C), indigenous ruderal competitors (RC), and alien competitors and introduced species (AC + I).

We have added the paragraph: In addition, in order to gain insights into community dynamics and ecological characteristics, we employed Social Behavior Types (SBT) for all plant species. SBT categorizes plant species based on their roles in the community and provides valuable information about community richness, stability, naturalness, niche occupancy, and response to disturbances or deviations from the natural state [38].” 

Reviewer comment: Lines 190-192: How were the bee identifications conducted, and by whom? The person who did the work should be named, and the taxonomic keys that were used to identify bees should be named and cited. (See Packer et al. 2018. Validating taxonomic identifications in entomological research. Insect Conservation and Diversity 11: 1-12 for more information.)

Milić et al.: Thank you for your comment; it was an oversight on our part. The author of this manuscript, Sonja Mudri-Stojnić, was responsible for identifying bee species and Tamara Tot for hoverflies. We have included the taxonomic keys that were used for bee species as well as for hoverflies identification.

Reviewer comment: Lines 234-236: It would be great to see a little more detail about what decision trees are in a general sense and what benefit the authors think they bring to this research.

Milić et al.: We added a paragraph in section Materials and methods discussing the general concept and benefits of the decision tree. 

Reviewer comment: Lines 260-262: It is not easy to see this point in Figure 2, especially for vegetation height (Fig. 2A). Can the authors revise this sentence to better summarize the figure?

Milić et al.: We accept the Reviewer's remark. We have revised the sentence and also made correction on Figure 2. 

Reviewer comment: Figure 2: It’s very difficult to tell which parts of the figure the uppercase letters A, B, and C are referring to. Can the authors make this clearer? Are the uppercase letters just pointing out that management type was significant across sites in their ANOVA analysis?

Milić et al.: Yes, uppercase letters indicate significant differences between grazing, mowing and control (different management practice). We also have made clearer Figure 2. 

Reviewer comment: Lines 282-283: Some of these numbers don’t make sense: “four species with >10 and 103 individuals”

Milić et al.: corrected. We have changed the sentence: “Most of the species had a very low abundance, with 25 species having 10 or fewer individuals, five species having 10 to 30 individuals, and only one species having over 100 individuals.” 

Reviewer #2: Exploring the effects of habitat management on grassland biodiversity: A case study from northern Serbia. 

This an interesting study presenting the effect of grassland management (mowing and grazing) on different biodiversity taxa. Although the study is valuable, it could be much improved. Some parts of the manuscript are not easy to follow, especially the introduction, due to poor writing. And the discussion section is too lengthy, should be shortened and more focused to the main questions assessed in the study. But the main concern is the sampling design, which is unbalanced (only 1 control site vs. 3 grazing and 3 mowing sites) and this should be acknowledged and dealt with proper statistical analysis in the study.

Milić et al.: We made some modifications to the Introduction section and condensed the discussion, aligning it more closely with the questions addressed in the study based on the reviewer's suggestions. Additionally, we included a paragraph detailing the statistical analysis employed to address the issue of imbalance in study sites.

Minor comments

Reviewer comment: L81: I recommend changing the sentence to make it clearer. By “initiatives” you refer to the conservation strategies? Something like “these conservation initiatives cover ecosystem services such as biomass production and…”, as you are referring here to the ecosystem services grasslands provide

Milić et al.: We have changed the sentence according Reviewer’s recommendation. 

Reviewer comment: L84 To what species you are referring to by “as many species are presently threatened”? Of solitary bees

Milić et al.: Yes, we meant on solitary bees. We have changed the sentence to make it clearer.

Reviewer comment: L83-87 the whole sentence would benefit from some rephrasing

Milić et al.: We have rephrased whole sentence.

Reviewer comment: L86. Fix typo, forging should be “foraging”

Milić et al.: corrected

Reviewer comment: L167: fix typo, nearest tree (NOT three)

Milić et al.: corrected

Reviewer comment: L171-175: treatments could be explained in a more structured and clear way, without mixing information

Milić et al.: We have restructured and clarified the presentation in this paragraph, separating the information more distinctly.

Reviewer comment: L186-192: no information on time of the day when pollinator sampling was conducted, please add

Milić et al.: We added information about the time of day and weather conditions during which the transect walks were conducted.

Reviewer comment: L205: I suggest rephrasing, species diversity was "calculated" instead of interpreted.

Milić et al.: corrected

Reviewer comment: L249: In the study area is not needed, I suggest to delete that

Milić et al.: corrected

Reviewer comment: L250. Please rephrase. Recorded instead of “noted presence”

Milić et al.: corrected

Reviewer comment: L253: Please correct. Unmanaged site, singular as there was only one site.

Milić et al.: corrected

Reviewer comment: Table 1. Are those numbers mean % across sites? Please specify. Also, would be better if the description of social behaviour types in table legend text follows the order of appearance of the table, starting with S, D, DT, NP and IAS

Milić et al.: We accept the Reviewer's remark. We have changed the title of the table to be more specified and have changed legend text follows the order of appearance of the table.

Reviewer comment: Table 4. Fix typo, distance to nearest TREE

Milić et al.: corrected

Reviewer comment: Fig. 3. IT would help to add the full name of management type in the legend in the figure: Control, Grazing, and Mowing. Also Variat should be replaced by Management.

Milić et al.: In Figure 3 we have added the full name of the management practice and replaced 'Variat' with 'Management' according Reviewer’s suggestion. 

Reviewer comment: Discussion could be shortened in order to improve flow and focus on main messages.

Milić et al.: We made the Discussion section shortened, focusing on the main messages in our research. We hope that we have successfully conveyed our key findings and insights.

---

## [Decision Letter · Decision Letter 1]

14 Mar 2024

Exploring the effects of habitat management on grassland biodiversity: A case study from northern Serbia

PONE-D-23-26216R1

Dear Dr. Milić,

We’re pleased to inform you that your manuscript has been judged scientifically suitable for publication and will be formally accepted for publication once it meets all outstanding technical requirements.

Kind regards,

Tzen-Yuh Chiang

Academic Editor

PLOS ONE

Additional Editor Comments (optional):

Reviewers' comments:

Reviewer's Responses to Questions

**Comments to the Author**

1. If the authors have adequately addressed your comments raised in a previous round of review and you feel that this manuscript is now acceptable for publication, you may indicate that here to bypass the “Comments to the Author” section, enter your conflict of interest statement in the “Confidential to Editor” section, and submit your "Accept" recommendation.

Reviewer #1: All comments have been addressed

Reviewer #2: All comments have been addressed

2. Is the manuscript technically sound, and do the data support the conclusions?

Reviewer #1: Yes

Reviewer #2: Yes

3. Has the statistical analysis been performed appropriately and rigorously? 

Reviewer #1: Yes

Reviewer #2: Yes

4. Have the authors made all data underlying the findings in their manuscript fully available?

Reviewer #1: (No Response)

Reviewer #2: Yes

5. Is the manuscript presented in an intelligible fashion and written in standard English?

Reviewer #1: Yes

Reviewer #2: Yes

6. Review Comments to the Author

Reviewer #1: The authors have addressed my comments on the original manuscript, and I commend them for their attention to detail in responding to my comments and the comments of the other reviewer.

Reviewer #2: I thank the authors for addressing the comments raised, although I would appreciate that they would have made reference to the specific lines where changes have taken place in the new version of the manuscript. Yet, having the tracked changes version at the end helped.

I think the current manuscript has improved substantially and I recommend it for publication. I particularly appreciate the additional information regarding on the statistical methods used to account for the imbalance study design.

I have only 2 minor comments on the current version:

- Lines 299-302. Please rephrase as current form it's not clear. Something similar to:

“Greatest number of plant, pollinator and bird species was found at the grazed (G) and mowed sites (M) with 85, 14, and 26 species and 115, 21, and 19 species respectively, while at unmanaged… “

- Conclusions. I would suggest to modify slightly the conclusions to strengthen the main findings of the work. What do you mean by “clear differentiation”? Also the role of mowing, in addition to grazing, could be highlighted as for example the highest number of Asteraceae and Fabaceae plant species, together with highest number of pollinators, were recorded in mowed sites.

Lines 651-653 are not very clear, I suggest rephrasing. For instance, when you say “modern management measures should follow this practice…“ which practice do you refer to? grazing with ruminants?

7. PLOS authors have the option to publish the peer review history of their article (what does this mean?). If published, this will include your full peer review and any attached files.

Reviewer #1: No

Reviewer #2: No

---

## [Editor Report · Acceptance letter]

20 Mar 2024

PONE-D-23-26216R1 

PLOS ONE

Dear Dr. Milić, 

I'm pleased to inform you that your manuscript has been deemed suitable for publication in PLOS ONE. Congratulations! Your manuscript is now being handed over to our production team.

Kind regards, 

on behalf of

Dr. Tzen-Yuh Chiang 

Academic Editor

PLOS ONE